# Potential Predictive Biomarkers of Systemic Drug Therapy for Hepatocellular Carcinoma: Anticipated Usefulness in Clinical Practice

**DOI:** 10.3390/cancers15174345

**Published:** 2023-08-30

**Authors:** Kenta Motomura, Akifumi Kuwano, Kosuke Tanaka, Yuta Koga, Akihide Masumoto, Masayoshi Yada

**Affiliations:** Department of Hepatology, Iizuka Hospital, 3-83 Yoshio-machi, Iizuka, Fukuoka 820-8505, Japan; akuwanoh1@aih-net.com (A.K.); ktanakah8@aih-net.com (K.T.); ykogah13@aih-net.com (Y.K.); amasumotoh1@aih-net.com (A.M.); myadah1@aih-net.com (M.Y.)

**Keywords:** hepatocellular carcinoma, tyrosine kinase inhibitors, immune checkpoint inhibitors, biomarkers

## Abstract

**Simple Summary:**

A number of agents, including immune checkpoint inhibitors, have become available for the treatment of hepatocellular carcinoma, but the objective response rate of these drugs is currently only 30% to 40%. Therefore, the identification of new predictive biomarkers and an increased knowledge of the mechanisms of response or resistance to systemic chemotherapies are required.

**Abstract:**

In the systemic drug treatment of hepatocellular carcinoma, only the tyrosine kinase inhibitor (TKI) sorafenib was available for a period. This was followed by the development of regorafenib as a second-line treatment after sorafenib, and then lenvatinib, a new TKI, proved non-inferiority to sorafenib and became available as a first-line treatment. Subsequently, cabozantinib, another TKI, was introduced as a second-line treatment, along with ramucirumab, the only drug proven to be predictive of therapeutic efficacy when AFP levels are >400 ng/mL. It is an anti-VEGF receptor antibody. More recently, immune checkpoint inhibitors have become the mainstay of systemic therapy and can now be used as a first-line standard treatment for HCC. However, the objective response rate for these drugs is currently only 30% to 40%, and there is a high incidence of side effects. Additionally, there are no practical biomarkers to predict their therapeutic effects. Therefore, this review provides an overview of extensive research conducted on potential HCC biomarkers from blood, tissue, or imaging information that can be used in practice to predict the therapeutic efficacy of systemic therapy before its initiation.

## 1. Introduction

Liver cancer is a major cause of death worldwide, and the number of people diagnosed with liver cancer is expected to increase [1]. As the principal histologic type of liver cancer, hepatocellular carcinoma (HCC) is responsible for the majority (approximately 75%) of liver cancer diagnoses and deaths [2]. Treatment options for localized HCC, such as surgical resection, ablation, liver transplantation, and transarterial chemoembolization, were established in the 20th century, but effective drug therapy for advanced HCC did not emerge until 2007 [3]. Although many clinical trials of potential drug therapies for unresectable HCC were conducted before the introduction of tyrosine kinase inhibitors (TKIs), no chemotherapeutic drugs demonstrated any significant survival benefit, as shown in the meta-analysis by Mathurin et al. [4]. Subsequently, an advanced understanding of the mechanisms of tumor cell proliferation and angiogenesis supported the development of the TKI, sorafenib [5]. In the Sorafenib HCC Assessment Randomized Protocol (SHARP) clinical trial of unresectable HCC, sorafenib showed a clear survival benefit over placebo and became the standard treatment for unresectable HCC in 2007 [6]. Subsequently, the results of the RESORSE trial, a clinical trial limited to unresectable HCC patients who tolerated sorafenib, led to the approval of regorafenib as a second-line therapy after sorafenib treatment in 2017 [7]. In 2018, lenvatinib was demonstrated to be non-inferior to sorafenib in the REFLECT trial, leading to a choice of sorafenib or lenvatinib as the first-line therapy [8]. In 2019, based on the results of the REACH-2 trial of cases with alpha-fetoprotein (AFP) levels of 400 ng/mL or more after sorafenib treatment, ramucirumab (an anti-vascular endothelial growth factor (VEGF) receptor antibody) became available as a second-line therapy [9]. Additionally, cabozantinib, a TKI, was developed as a second-line or later therapy [10].

In the following years, immune checkpoint inhibitors (ICIs) emerged. In 2017, nivolumab demonstrated promising results in the Checkmate040 phase I/II trial [11], leading to its approval by the Food and Drug Administration. Additionally, pembrolizumab was approved by the Food and Drug Administration as a first-line and second-line treatment based on the results of the phase II KEYNOTE-224 [12] and phase III KEYNOTE-240 trials [13].

In 2020, a combination therapy of atezolizumab and bevacizumab, which surpassed sorafenib in clinical trials, was introduced [14]. As a result, both sorafenib and lenvatinib were relegated to second-line or later therapies. Furthermore, a combination therapy of two ICIs, durvalumab and tremelimumab (STRIDE regimen), outperformed sorafenib in the treatment results of the HIMALAYA trial. Durvalumab monotherapy also showed non-inferiority to sorafenib [15], allowing these therapeutics to be added as new treatment options.

Through these trials, a number of agents, including ICIs, have become available for the treatment of HCC. However, the objective response rate (ORR) of these drugs is currently only 30% to 40%, with a high incidence of side effects [6,7,8,9,10,11,12,13,14,15]. Other than ramucirumab, which requires an AFP level of 400 ng/mL or more, there are no practical biomarkers to predict the therapeutic effects of these treatment methods.

Most of the systemic therapies for HCC are administered in general hospitals without research facilities. In such hospitals, imaging tests such as computed tomography and magnetic resonance imaging (MRI), and pathological diagnoses, including tumor tissue sampling and immunohistochemistry (IHC) staining, can be conducted. However, the introduction of advanced technologies, such as the analysis of tumor genomic or transcriptomic profiles, miRNA evaluation, cell cultures, and the identification of driver gene mutations, is often challenging. These limitations arise due to constraints in facilities, personnel, and financial implications.

In this review, we have provided an overview of potential biomarkers that, with further prospective validation, could be translated into clinical applications in the future. The aim of our review is to offer readers a comprehensive perspective, enabling them to assess which of these potential biomarkers might be feasibly implemented within their specific hospital environments.

## 2. Exploring Potential Biomarkers to Predict the Therapeutic Effects of TKIs (Table 1)

### 2.1. Potential Biomarkers for Sorafenib

The first TKI, sorafenib, has remained the first drug of choice for a decade, leading to numerous early studies being conducted on biomarkers to predict its therapeutic effect.

One of the objectives of the phase III SHARP trial was to investigate plasma biomarkers to predict prognoses and therapeutic effects, with 10 plasma biomarkers measured at baseline and after 12 weeks of treatment [16]. As a result, baseline angiopoietin 2 (Ang-2) and VEGF concentrations were found to be independent predictors of survival in the entire advanced HCC patient population. However, these were not unique to the sorafenib cohort and were similar to the placebo cohort [16].

In a real-world study in Japan, Miyahara et al. measured the serum levels of eight pro-angiogenic cytokines (Ang-2, follistatin (FST), granulocyte colony-stimulating factor (G-CSF), hepatocyte growth factor (HGF), leptin, platelet-derived growth factor-BB (PDGF-BB), platelet endothelial cell adhesion molecule-1 (PECAM-1), and VEGF) in 120 consecutive HCC patients treated with sorafenib. They reported that high expression of Ang-2 or three or more pro-angiogenic cytokines was associated with poor progression-free survival (PFS) and overall survival (OS) in patients treated with sorafenib [17]. In addition, the presence of macrovascular invasion (MVI) was also shown to be associated with poor OS related to clinical parameters [17].

In an analysis of pooled data from the SHARP and Asia Pacific (AP) phase III trials, a significantly greater OS benefit compared with placebo was observed in patients without extrahepatic spread (EHS; hazard ratio (HR), 0.55 vs. 0.84), with hepatitis C virus (HCV) (HR, 0.47 vs. 0.81), and a low neutrophil-to-lymphocyte ratio (NLR) (HR, 0.59 vs. 0.84) [18]. In this analysis, the NLR was divided into >3.0 and <3.0, which was the median in the sorafenib administration group. Although this was a retrospective study, it described certain factors that had significant differences compared with the placebo group. However, these findings were strictly predictions of the survival period and not of drug effectiveness.

The NLR reflects the inflammatory response to cancer, and its elevation is recognized as an indicator of poor prognosis [19,20]. Qi et al. conducted a systematic review and meta-analysis of 20,475 HCC patients from 90 articles to explore the prognostic role of NLR in HCC and reported that a lower baseline NLR was significantly associated with the survival period (HR, 1.80, 95% confidence interval (CI): 1.59–2.04, *p* < 0.00001) [21]. This study also conducted a subgroup meta-analysis for cases treated with sorafenib, and although it was not compared with a placebo group, a low NLR was associated with better survival in 170 cases between high and low NLR groups [21].

The GIDEON trial, a large prospective observational registration study focused on assessing the safety of sorafenib treatment in the real world, showed better OS in Child–Pugh A patients compared with Child–Pugh B patients. Moreover, a univariate Cox regression analysis of each factor of the Child–Pugh score showed that albumin and bilirubin, which form the ALBI score, strongly influenced OS [22]. The authors did not utilize anything other than descriptive statistics that considered the impact of selection bias. Additionally, this study lacked a control group and was not randomized. However, liver functional reserve can clearly affect the patient’s survival period after treatment with sorafenib for HCC. For example, several studies have shown that in advanced HCC cases related to HCV, OS is extended as the liver functional reserve improves following HCV eradication with interferon (IFN) or direct-acting antivirals [23,24].

For pathological biomarkers in tumor tissues, phase II trial results indicated that phosphorylated ERK might be a useful biomarker to predict the prognosis of patients treated with sorafenib. This protein is located downstream of Raf kinase in the MAPK cascade, which is a major target of sorafenib [25]. Although there have been studies in clinical settings that have shown favorable efficacy [26], several have also shown unfavorable efficacy [27,28]. Therefore, no consensus has been reached regarding phosphorylated ERK.

Arao et al. reported that in 13 cases with significant tumor shrinkage after sorafenib treatment, fibroblast growth factor (FGF)3/FGF4 amplification was observed in the tumor genome. Additionally, multiple lung metastases in poorly differentiated histological types were seen as clinical pathological features. Although the sample size was relatively small, FGF3/FGF4-amplified tumors were frequently observed in responders to sorafenib [29]. Although this study examined biomarkers for sorafenib treatment efficacy, no further research with additional cases to support these findings was conducted.

Tumor tissues can also be used to investigate microRNA (miRNA) expression. miRNAs are small endogenous non-coding RNAs that inhibit translation or support cleavage of mRNAs to negatively regulate gene expression. These molecules are highly stable and can be reliably detected in stored clinical samples and cell cytology specimens, making them ideal biomarker candidates [30,31,32]. Various miRNAs are also mechanistically involved in the development, proliferation, and progression of HCC and can be detected in serum and plasma samples, suggesting they might be used as diagnostic markers [33].

Gyöngyösi et al. investigated the expression levels of 14 miRNAs in 20 HCC cases where tumor tissue samples were collected by fine needle aspiration before sorafenib administration, with the data demonstrating the high expression of miR-224 was associated with increased PFS and OS rates [31].

Vaira et al. conducted a comprehensive profiling of approximately 700 miRNAs in a series of 26 HCC patients treated with sorafenib (training set) using tumor tissues collected prior to treatment, then verified the results in an independent series of 58 patients (validation set) [34]. As a result, six miRNAs were found to be significantly associated with clinical variables in the training set. Of these, only miR-425-3p was significant in the validation set, with high miR-425-3p levels being associated with a longer time to progression (TTP) and PFS [34]. However, no follow-up studies have been conducted to date. In general, miRNA-related cancer research is primarily focused on the development of therapies that target specific miRNAs or the use of these molecules as a tool for early cancer detection [35].

Other studies have constructed high-throughput assay systems in completely different ways. Qiu et al. created a Liver Cancer Model Repository (LIMORE) panel of 81 cell lines by creating 50 patient-derived liver cancer cell lines, in addition to 31 existing cell lines, to model HCC heterogeneity. The authors examined the sensitivity of these cells to a total of 90 drugs. By using this panel, which has verified gene mutations and gene expression characteristics, it is possible to identify gene–drug interactions of therapeutic methods and biomarker candidates. When predicting the effect of sorafenib treatment, Dickkopf-1 (DKK-1) was identified as a potentially useful biomarker [36]. Interestingly, DKK-1 is a secreted protein that antagonizes Wnt signal transduction, which is known to affect ICI efficacy [37]. Because DKK-1 is a serum protein, it may be relatively easy to verify its potential as a biomarker in existing cohorts with preserved serum samples.

### 2.2. Potential Biomarkers for Regorafenib

Although insufficient results were obtained from the biomarker studies of the phase III SHARP trial for sorafenib, a more comprehensive exploratory biomarker analysis was conducted for patients in the RESORCE trial at the DNA, RNA, and protein levels [38]. Of the 266 proteins studied in baseline plasma samples, decreases in five, Ang-1, cystatin B, the latency-associated peptide of transforming growth factor beta 1 (LAP TGF-β1), oxidized low-density lipoprotein receptor 1 (LOX-1), and C-C motif chemokine ligand 3 (MIP-1α), were found to be associated with extended TTP and OS. Moreover, nine plasma miRNAs, miR-30a, miR-122, miR-125b, miR-200a, miR-374b, miR-15b, miR-107, miR-320, and miR-645, were related to OS, although none were associated with TTP. Furthermore, there was no apparent correlation between the AFP or c-MET protein expression levels and the OS or TTP benefits of regorafenib treatment, causing them to be excluded as potential predictive biomarkers [38]. Currently, with treatments including ICIs becoming the standard of care as first-line treatments, the number of cases where regorafenib is used after sorafenib treatment is likely to be low. This could make planning prospective validation studies challenging. However, as these potential biomarkers are plasma proteins and miRNAs, it is hoped that they can be validated in existing cohorts using blood samples that have been preserved.

### 2.3. Signaling Pathways as Biomarkers for TKIs: Insights from Trials with mTOR and MET Inhibitors

In several cancer types, other than HCC, driver gene mutations in signaling pathways strongly promote tumor growth. This has led to established biomarker-driven treatment concepts for drug selection and predicting treatment outcomes [39,40,41]. In HCC, signaling pathways such as RAS, mammalian target of rapamycin (mTOR), MET, and FGF-19 have been considered potential therapeutic targets. Several clinical trials were conducted using inhibitors targeting these pathways, but unfortunately, many ended with disappointing results [42]. For example, mTOR signaling is activated in about half of all HCC cases and is associated with worse outcomes [43]. Despite this strong theoretical basis for using the mTOR inhibitor everolimus to treat HCC, the final results from a phase III trial did not suggest any trend of prolonged OS (everolimus vs. placebo, 7.6 vs. 7.3 months) [44].

The potential cause of failure in these clinical trials may be the inclusion of all patients with unresectable HCC. It is considered desirable to incorporate molecular selection factors into prospective research, as in clinical trials for other cancers [45]. One of the drugs for which there was hope for biomarker-driven treatment in HCC was tivantinib, an MET inhibitor. In a phase II trial against placebo as a second-line treatment, tivantinib improved survival rates in patients with high tumor MET expression levels although no significant effect was observed in all cases [46]. From these results, phase III trials (METIV-HCC, JET-HCC) were conducted comparing tivantinib and placebo only in patients with high MET expression [47,48]. However, in both trials, no statistically significant treatment effect was observed for tivantinib compared with placebo.

Cabozantinib, which targets several TKs including MET, VEGF, and AXL, was successful in a phase III trial (CELESTIAL) [10]. In this trial, baseline plasma levels of MET, AXL, VEGFR2, HGF, GAS6, VEGF-A, PlGF, IL-8, EPO, ANG2, IGF-1, VEGF-C, and c-KIT were evaluated as biomarkers; however, none of these predicted the treatment effect of cabozantinib on OS or PFS [49].

The cause of HCC is diverse, including viral infection, toxin exposure, and metabolic disorders. From the results of large-scale genomic analyses, it has become clear that gene mutations in HCC are centered on diverse non-drug targetable mutations, such as TERT, CTNNB1, and TP53 [50,51]. Furthermore, HCC is heterogeneous even within an individual patient, and sequencing analysis of a single lesion cannot fully characterize the genomic features of HCC in certain cases [52]. In clinical trials of tyrosine kinase inhibitors, the failure to successfully use the expression of specific therapeutic target molecules or the activation of signaling pathways as biomarkers suggests that hepatocellular carcinoma may have low dependency on these signaling pathways, reflecting its underlying heterogeneity and diversity [53].

### 2.4. Potential Biomarkers for Lenvatinib

Lenvatinib is a multi-kinase inhibitor that inhibits vascular endothelial growth factor receptors (VEGFR) 1–3, fibroblast growth factor receptors (FGFR) 1–4, platelet-derived growth factor receptor (PDGFR) α, and oncogenes RET and KIT [54]. Preclinical studies have shown that lenvatinib has strong anti-angiogenic activity, primarily through inhibition of the VEGF and FGF signaling pathways [55].

In a subgroup analysis of the REFLECT trial, patients with HBV infection or alcohol as underlying factors showed better PFS rates with lenvatinib treatment than with sorafenib [8]. However, no biomarker exploration beyond the subgroup analysis was planned in the REFLECT trial.

Tada et al. focused on the associations between outcomes in HCC patients treated with lenvatinib and the NLR. In a multivariate analysis of a cohort of 237 individuals, an NLR ≥ 4 was independently associated with OS and PFS. There was also a significant difference in the disease control rate between patients with low NLR (<4) and high NLR (≥4) (85.5% vs. 67.3%, *p* = 0.007). A spline curve analysis showed that an NLR of approximately 3.0 to 4.5 was an appropriate cutoff value related to OS [56]. In addition, in the retrospective RELEVANT study from 23 other facilities, data were collected for 1325 patients treated with lenvatinib. In the multivariate analysis of OS, HBsAg positivity, NLR > 3, and AST > 38 were independently associated with poor prognosis in all three groups. Furthermore, nonalcoholic fatty liver disease (NAFLD)/nonalcoholic steatohepatitis (NASH)-related etiology was independently associated with a good prognosis. The multivariate analysis showed that NAFLD/NASH, Barcelona Clinic Liver Cance (BCLC) stage, NLR, and AST were independent prognostic factors for PFS in cases treated with lenvatinib [57]. However, these studies did not make comparisons with placebo or other drug treatments. Of note, a control group or placebo group is necessary to identify predictors of the therapeutic effect of a certain drug; thus, it is necessary to consider the limitations of such single-arm observational studies when evaluating predictive markers.

Qiu et al., who identified DKK-1 as a potential predictor of sorafenib effectiveness, used a panel of 81 HCC cell lines from the Liver Cancer Model Repository to show that FGFR inhibitors including lenvatinib had a favorable effect on HCC strains with the amplification of FGFR and FGF. They suggested that the amplification of FGF19 and FGFR might be biomarkers for lenvatinib effectiveness [36].

Myojin et al. developed a new HCC mouse model that reproduced the diversity of tumor driver genes by introducing a pooled cancer gene cDNA library using transposon-based intrahepatic delivery. This could be used to simultaneously evaluate the individual effects of various genetic drivers on TKI sensitivity in HCC in vivo. This model revealed that tumors expressing FGF19 were sensitive to lenvatinib in vivo. They comprehensively evaluated tumor secretory proteins to discover biomarkers for FGF19-driven HCC, identifying a correlation between FGF19 and the secretory protein ST6 β-galactoside α-2,6-sialyltransferase 1 (ST6GAL1) in HCC cells. This provided clinical evidence that ST6GAL1 may be a useful serum biomarker for the selection of HCC patients who may derive more benefit from lenvatinib than sorafenib treatment [58]. This study strongly focused on the exploration of biomarkers for the therapeutic effect of lenvatinib in cancer cells, but validation in a prospective cohort is necessary for clinical application. Because ST6GAL1 protein levels can be measured in serum samples, it would be a very useful biomarker if validated.

Lenvatinib was reported to have a higher selectivity for FGFR compared with other kinase inhibitors [54,59]. Therefore, biomarkers related to FGF-FGFR signaling may be more promising than those related to other signaling pathways that have been previously studied.

**Table 1 cancers-15-04345-t001:** Factors influencing patient prognosis or efficacy when treating hepatocellular carcinoma with tyrosine kinase inhibitors.

Therapeutics	Study Design	Number of Cases	Prognostic and Predictive Factors	Outcome	Statistical Analysis HR (95% CI)	*p*-Value	Authors [Reference No.]
Sorafenib	Retrospective, single-arm	120	High serum Ang-2	PFS ↓	Univariate 1.84 (1.21–2.81)	0.004	Miyahara K et al. [17]
				OS ↓	Multivariate 1.83 (1.12–2.98)	0.014	
			High angiogenic group * *: patients with >three serum cytokines (Ang-2, FST, G-CSF, HGF, Leptin, PDGF-BB, PECAM-1, or VEGF)	PFS ↓	Univariate 1.98 (1.30–3.06)	0.001	
				OS ↓	Multivariate 1.76 (1.07–2.94)	0.023	
			MVI (present)	OS ↓	Multivariate 2.27 (1.36–3.72)	0.001	
Sorafenib	Retrospective pooled analysis of two phase 3 trials (vs. placebo)	Sorafenib 448 Placebo 379	Without EHS	OS ↑	Multivariate 0.55 (0.42–0.72)	0.015	Bruix J et al. [18]
			With HCV	OS ↑	Multivariate 0.47 (0.32–0.69)	0.035	
			Low NLR	OS ↑	Multivariate 0.59 (0.46–0.77)	0.0497	
Sorafenib	Subgroup meta-analyses, single-arm	170	Low NLR	OS ↑	Univariate 1.49 (1.17–1.91)	0.001	Qi X et al. [20]
Sorafenib	Observational registry, single-arm	3371	Child–Pugh A	OS ↑	Kaplan–Meier	N/A	Marrero JA et al. [22]
			Bilirubin	OS	Univariate 1.71 (1.57–1.86)	N/A	
			Albumin	OS	Univariate 1.76 (1.63–1.89)	N/A	
Sorafenib	Retrospective, single-arm, HCV patients only	103	HCV eradication	OS ↑	Multivariate 0.46 (0.26–0.78)	0.004	Kuwano A et al. [23]
			ALBI score	OS	Multivariate 2.29 (1.20–4.37)	0.012	
Sorafenib	Population-based retrospective cohort, HCV patients only, single-arm	1684	DAA user	OS ↑	Univariate PSM univariate	<0.0001 <0.0001	Tsai H-Y et al. [24]
Sorafenib	Retrospective, single-arm	55	FGF3/FGF4 amplification (Frozen tumor tissue)	CR/PR ↑	Fisher’s exact	0.006	Arao T et al. [29]
			Multiple lung metastases	CR/PR ↑	Fisher’s exact	0.006	
Sorafenib	Retrospective, single-arm	20	High miR-224 expression (FFPE tumor tissue)	PFS ↑	Univariate 0.28 (0.09–0.92)	0.029	Gyöngyösi B et al. [31]
				OS ↑	Univariate 0.24 (0.07–0.79)	0.012	
Sorafenib	Retrospective, single-arm	Training 26 Validation 58	High miR-425-3p expression (FFPE tumor tissue)	TTP ↑	Multivariate 0.4 (0.1–0.7)	0.002	Vaira V et al. [34]
				PFS ↑	Multivariate 0.3 (0.1–0.7)	0.0012	
Sorafenib	Retrospective validation of the pharmacogenomics panel, single-arm	54	High serum DKK-1	PFS ↑	Univariate	0.0396	Qiu Z et al. [36]
				OS ↑	Univariate	0.0171	
Regorafenib	Retrospective pooled analysis of the phase 3 trial (vs. placebo)	Protein cohort Regorafenib 332 Placebo 167	Plasma ANG-1 (1 ng/mL increase)	OS ↓	Multivariate 1.12 (1.05–1.19)	0.019	Teufel M et al. [38]
				TTP ↓	Multivariate 1.10 (1.04–1.17)	0.017	
			Low plasma Cystatin-B (2-fold increase)	OS ↓	Multivariate 1.46 (1.15–1.85)	0.04	
				TTP ↓	Multivariate 1.42 (1.14–1.77)	0.018	
			Low plasma LAP TGF-β1 (2-fold increase)	OS ↓	Multivariate 1.36 (1.12–1.65)	0.04	
				TTP ↓	Multivariate 1.41 (1.18–1.68)	0.004	
			Low plasma LOX-1 (1 ng/mL increase)	OS ↓	Multivariate 1.35 (1.16–1.57)	0.009	
				TTP ↓	Multivariate 1.78 (1.33–2.39)	0.003	
			Low plasma MIP-1α (1 pg/mL increase)	OS ↓	Multivariate 1.02 (1.01–1.04)	0.04	
				TTP ↓	Multivariate 1.02 (1.00–1.03)	0.043	
		miRNA cohort Regorafenib 234 Placebo 109	Decreased miR-15b	OS ↑	Multivariate 0.37 (0.20–0.70)	0.002	
			Decreased miR-107	OS ↑	Multivariate 0.54 (0.37–0.81)	0.003	
			Decreased miR-320b	OS ↑	Multivariate 0.57 (0.41–0.81)	0.001	
			Increased miR-122	OS ↑	Multivariate 1.35 (1.14–1.60)	0.0004	
			Increased miR-374b	OS ↑	Multivariate 1.36 (1.11–1.65)	0.002	
			Increased miR-200a	OS ↑	Multivariate 1.39 (1.15–1.68)	0.001	
			Increased miR-30a	OS ↑	Multivariate 1.47 (1.14–1.88)	0.003	
			Increased miR-125b	OS ↑	Multivariate 1.54 (1.19–1.99)	0.001	
			Absence miR-645 * (* dichotomized analysis, not vs. placebo)	OS ↑	Multivariate 3.16 (1.52–6.55)	0.002	
Lenvatinib	Subgroup analysis of the open-label phase 3 trial (vs. sorafenib)	Lenvatinib 478 (HBV 251, Alcohol 36) sorafenib 476 (HBV 228, Alcohol 21)	HBV	PFS ↑	Univariate 0.62 (0.50–0.75)	N/A	Kudo M et al. [8]
			Alcohol	PFS ↑	Univariate 0.27 (0.11–0.66)	N/A	
Lenvatinib	Retrospective, single-arm	237	NLR ≥ 4	OS ↓	Multivariate 1.87 (1.10–3.12)	0.021	Tada T et al. [56]
				PFS ↓	Multivariate 1.90 (1.27–2.84)	0.002	
				DCR ↓	Chi-square test?	0.007	
			AFP ≥ 400 ng/mL	OS ↓	Multivariate 1.97 (1.19–3.27)	0.009	
			mALBI grade 2b or 3	OS ↓	Multivariate 2.12 (1.27–3.56)	0.004	
			BCLC stage ≥ C	PFS ↓	Multivariate 1.52 (1.03–2.24)	0.036	
Lenvatinib	Retrospective, single-arm	1325	HBV	OS ↓	Multivariate 1.56 (1.13–2.17) *	0.0071 *	Casadei-Gardini A et al. [57] *: Data are from the model 1 of 3 multivariate analyses.
			NAFLD/NASH	OS ↑	Multivariate 0.58 (0.33–0.98) *	0.0044 *	
				PFS ↑	Multivariate 0.87 (0.75–0.93)	0.0090	
			BCLC stage C	OS ↓	Multivariate 1.64 (1.19–2.27) *	0.0027 *	
				PFS ↓	Multivariate 1.33 (1.14–1.55)	0.0002	
			NLR > 3	OS ↓	Multivariate 1.95 (1.46–2.60) *	<0.0001 *	
				PFS ↓	Multivariate 1.16 (1.01–1.36)	0.0482	
			AST > 38	OS ↓	Multivariate 1.52 (1.08–2.13) *	0.0167 *	
				PFS ↓	Multivariate 1.21 (1.01–1.45)	0.0365	
Lenvatinib	Retrospective validation of the experimentally identified biomarker (vs. sorafenib)	Lenvatinib 65 (ST6GAL1 high 22, low 43) sorafenib 31 (ST6GAL1 high 12, low 19)	Serum ST6GAL1 high	OS ↑	Univariate	<0.05	Myojin Y et al. [58]

HR, hazard ratio; CI, confidence interval; PFS, progression-free survival; OS, overall survival; CR, complete response; PR, partial response; TTP, time to progression; DCR, disease control rate; HBV, hepatitis B virus; HCV, hepatitis C virus; MVI, macrovascular invasion; EHS, extrahepatic spread; NLR, neutrophil to lymphocyte ratio. Note: In the “Statistical analysis” section, “univariate” typically refers to the Kaplan–Meier method and log-rank test, and the inclusion of HR indicates the use of Cox regression. Additionally, ‘multivariate’ typically refers to the utilization of the multivariate Cox regression model.

## 3. AFP as an Approved Predictive Biomarker for Ramucirumab Treatment

VEGF and VEGFR2 signaling pathways have a crucial role in angiogenesis and tumor growth [60]. Multi-kinase inhibitors, such as sorafenib and lenvatinib, which have been shown to be effective against HCC, target VEGFR2. Ramucirumab is a human IgG1 monoclonal antibody that inhibits the ligand activation of VEGFR2 [61]. In a phase II trial of ramucirumab as a first-line therapy for HCC, ramucirumab demonstrated an ORR and OS that surpassed the sorafenib administration group in the SHARP trial [62]. In this trial, an exploratory study of biomarkers measured circulating VEGF, soluble VEGFR1 (sVEGFR1), sVEGFR2, and several cytokines and growth factors in serum samples after ramucirumab administration. Among them, a potential correlation was suggested between reduced serum sVEGFR1 levels until day 8 post-administration and prolonged PFS and OS [61]. However, in the REACH trial, a phase III trial of ramucirumab vs. placebo as a second-line therapy after sorafenib treatment, a significant improvement in OS was not achieved in the ramucirumab group compared with the placebo group [63].

Apart from the initially explored biomarkers, a subgroup analysis of the REACH trial revealed that OS in the ramucirumab group was significantly better when limited to cases with an AFP level ≥ 400 ng/mL [63]. Therefore, the REACH-2 trial was planned, which was a phase III trial of ramucirumab vs. placebo as a second-line therapy restricted to cases with an AFP level ≥ 400 ng/mL after sorafenib treatment [9]. As expected, the REACH-2 trial results indicated that OS was significantly extended in the ramucirumab treatment group compared with the placebo group. This trial became the first successful phase III trial for advanced HCC treatment that selected target cases using a biomarker [9].

Because AFP was shown to be a predictive biomarker for the therapeutic effect of ramucirumab, an analysis of 520 HCC cases with known baseline AFP values was conducted to investigate the molecular profile differences of tumors using AFP levels. The data suggested that tumors in cases with an AFP level > 400 ng/mL showed significant activation of VEGF signaling [64].

## 4. Exploration of Potential Biomarkers to Predict the Therapeutic Efficacy of Single-Agent ICIs and Combined Immunotherapy (Table 2)

The pharmacotherapy of HCC has shifted from being dominated by TKIs to ICIs and combined immunotherapies. Correspondingly, research into biomarkers to predict therapeutic effectiveness has transitioned from focusing on those related to tumor growth signals to those focusing on the tumor microenvironment and tumor immune environment [65].

Studies of biomarkers to predict the therapeutic effects of ICIs for HCC began with the validation of biomarkers discovered in other cancer types, such as melanoma, non-small cell lung cancer, and colorectal cancer. However, despite a demonstrated response rate of about 20% for HCC cases to single-agent therapies such as nivolumab and pembrolizumab, these treatments did not become standard first-line or second-line therapies [10,11,12,66,67]. The established combination therapy of atezolizumab and bevacizumab, which are anti-PD-L1 and anti-VEGF-A antibodies, respectively, became the standard treatment. This resulted in the research of biomarkers to predict the therapeutic effects of ICIs to focus on this type of combined immunotherapy.

### 4.1. Known Candidate Predictive Markers of the Efficacy of Single-Agent ICI and Combined Immunotherapies for HCC: PD-L1 Expression, Tumor Mutation Burden (TMB), and Microsatellite Instability (MSI)

The discovery of immune checkpoint proteins, such as PD-1/PD-L1 and CTLA-4, represents a significant breakthrough in the cancer immunotherapy field [68,69]. Currently, anti-PD-1 antibodies, anti-PD-L1 antibodies, and anti-CTLA-4 antibodies are used to treat HCC, including in combination therapies. For other types of cancer, PD-L1 expression, TMB, and MSI have been reported as biomarkers to predict the therapeutic effects of these ICIs [70]. However, when considering the practicality of these biomarkers, the frequency of PD-L1 expression, TMB-High, and MSI-High becomes an issue. According to a large cohort study by Ang et al., the incidence of MSI-High cases in HCC was extremely limited, with only one case among 542 patients. Additionally, only six cases (0.8%) among 755 cases had a TMB of 20 mutations/Mb or more [71].

Zhu et al. conducted comprehensive analyses of transcriptomics, genomics, and IHC staining of patient samples collected in the phase Ib GO30140 and phase III IMbrave150 trials to explore biomarkers for atezolizumab and bevacizumab combination therapy [72]. Whole exome sequencing (WES) or FoundationOne panel profiling was performed to evaluate TMB, resulting in median TMBs of 5.6 mutations/Mb and 4.4 mutations/Mb, respectively. TMB was categorized as low, medium, or high, and its associations with response rates and survival times were verified. However, no relationship between TMB and response rate or survival benefit was observed in the GO30140 trial arm A and IMbrave 150 [72].

PD-L1 expression merits further investigation. In the phase I/II CheckMate 040 trial of nivolumab monotherapy for advanced HCC, PD-L1 expression in tumor tissues was examined in 174 out of 214 cases in the dose-expansion phase. Of these, 34 cases (20%) showed PD-L1 expression ≥ 1% in tumor cells via IHC, and these cases demonstrated an ORR of 9/34 (26%; 95% CI 13–44). However, even in 140 cases with PD-L1 < 1%, an ORR was observed in 26/140 cases (19%; 95% CI 13–26), suggesting that therapeutic responses were observed regardless of PD-L1 expression status [11].

In the phase III Checkmate459 trial of nivolumab vs. sorafenib, PD-L1 expression ≥ 1% in tumor cells was found in 71 of 366 cases (19%) in the nivolumab group and 64 of 362 cases (18%) in the sorafenib group. In patients administered nivolumab, a higher ORR was indicated if they had PD-L1 ≥ 1% vs. PD-L1 < 1% (PD-L1 ≥ 1% ORR 20/71 (28%; 18–40); PD-L1 < 1% ORR 36/295 (12%; 9–17)). However, in the sorafenib group, there was no difference in ORR between those with PD-L1 ≥ 1% or PD-L1 < 1% (PD-L1 ≥ 1% ORR 6/64 (9%; 4–19); PD-L1 < 1% ORR 20/300 (7%; 4–10)). A comparison between those with PD-L1 ≥ 1% in the nivolumab group and those with PD-L1 ≥ 1% in the sorafenib group indicated a trend that favored nivolumab with a median OS of 16.1 months (95% CI 8.4–22.3) vs. 8.6 months (95% CI 5.7–16.3), but the difference was not statistically significant (HR 0.80 (0.54–1.19)) [66].

In the phase II Keynote224 trial of pembrolizumab monotherapy, the conventional positive cell rate of PD-L1 in tumor cells (tumor proportion score (TPS)) and the combined positive score (combined positive score (CPS)) were calculated by dividing the number of PD-L1 positive cells in tumor cells and immune cells by the total number of surviving tumor cells and multiplying by 100 [73]. Of 52 cases, 22 (42%) were CPS positive and only seven cases (13%) were TPS positive. Significant differences were observed in the response rates and PFS between CPS positive and negative cases, but not between TPS positive and negative cases [73].

According to the comprehensive analysis by Zhu et al., patients with high CD274 (PD-L1 mRNA) expression had a longer PFS with the atezolizumab–bevacizumab combination therapy than those with low expression. However, IHC data for PD-L1 protein levels indicated there was only a potential correlation between PD-L1 expression and response [72].

As these results suggest, PD-L1 expression is somewhat related to the efficacy of anti-PD-1/PD-L1 immunotherapy in HCC and might be a biomarker to predict therapeutic effects. However, there are some uncertainties in its benefit because it is difficult to definitively make these predictions at the protein level, whether to use TPS or CPS scoring has not been determined, and there is an issue of heterogeneity associated with IHC staining assays [74].

### 4.2. NASH as a Background Liver Disease

One potential biomarker that might predict the lack of efficacy of ICI monotherapy is NASH/NAFLD as a background liver disease. Pfister et al. suggested that in a mouse model of NASH-induced HCC, CD8+/PD-1+ T cells promoted the progression of NASH. The administration of ICIs “released the brakes” on these NASH-promoting cells, resulting in a potential exacerbation of NASH and increased HCC occurrence [75]. The authors conducted a meta-analysis of the cohorts from three phase III trials where ICIs were administered, namely Checkmate 459, IMbrave 150, and KEYNOTE-240. This analysis showed that although ICI treatment significantly prolonged OS compared with the control in HBV-related and HCV-related HCC cases, the prognosis did not improve in non-viral HCC cases. Furthermore, in two separate retrospective cohorts treated with anti-PD-1 or anti-PD-L1 antibodies, HCC cases caused by NAFLD showed reduced OS compared with those with other etiologies [75]. As demonstrated in this study, a subgroup analysis of the phase III IMbrave 150 trial of atezolizumab–bevacizumab combination therapy and sorafenib showed non-viral HCC including NASH did not show superiority, with a median OS of 17.0 months in the atezolizumab–bevacizumab combination group compared with 18.1 months in the sorafenib group [76].

Moreover, in a multicenter study involving 36 facilities in four countries (Italy, Japan, South Korea, and the UK), a retrospective analysis of 759 cases of advanced non-viral HCC revealed that when lenvatinib and atezolizumab + bevacizumab treatments were compared, lenvatinib had significantly better OS and PFS rates in non-viral HCC overall. When non-viral HCC was divided into NAFLD/NASH and non-NAFLD/NASH, lenvatinib treatment was associated with a significant survival benefit compared with atezolizumab + bevacizumab in patients with NAFLD/NASH HCC [77].

However, not all evidence suggests ICIs are less active in patients with a non-viral etiology. In a recent post hoc analysis of IMbrave150, clinical data from 279 out of 336 patients were obtained and analyzed, with etiologies categorized into HBV, HCV, alcohol, and NAFLD. However, no significant differences in OS, PFS, or ORR were observed between the different etiologies [78]. Etiology seems to have a role in modulating the response to ICIs, and currently, no recommendations regarding different treatments based on the underlying liver disease have been made.

### 4.3. Wnt/β-Catenin Mutations as a Biomarker and MRI Findings as Imaging Biomarkers

Spranger et al. reported the presence of Wnt/β-catenin mutations in melanoma resulted in the exclusion of T cell infiltration and resistance to ICIs [79]. Using The Cancer Genome Atlas (TCGA), Luke et al. demonstrated that tumors lacking the genetic expression signature of T cell-mediated inflammation, including 31 types of solid cancers including melanoma, had activated Wnt/β-catenin signaling [80]. Furthermore, Harding et al. conducted a genomic analysis of 127 HCC tumor tissues and reported that Wnt/β-catenin mutations were present in 45% of cases. Although the presence or absence of these mutations did not affect PFS with sorafenib treatment, their presence significantly shortened PFS with ICI treatment (2.0 vs. 7.4 months, *p* < 0.0001) [81]. In addition, in a comprehensive study by Zhu et al., patients with a wild-type CTNNBI genotype in the IMbrave150 trial showed a greater therapeutic effect with atezolizumab + bevacizumab compared with sorafenib treatment, but no significant difference was observed between the treatments in cases with CTNNBI mutations [72].

Ueno et al. focused on the differences in HCC findings in the hepatobiliary phase of gadolinium ethoxybenzyl diethylenetriamine pentaacetic acid (Gd-EOB-DTPA)-enhanced MRI, comprehensively examined the transporter of Gd-EOB-DTPA and analyzed the molecular regulatory mechanism. Using clinical samples, they demonstrated that high expression levels of OATP1B3 were strongly correlated with greater enhancement in the hepatobiliary phase of Gd-EOB-DTPA-enhanced MRI. Additionally, activated Wnt/β-catenin signaling was closely associated with OATP1B3 expression in HCC cell cultures [82].

Aoki et al. analyzed 18 HCC cases that had received anti-PD-1 or anti-PD-L1 monotherapy and had Gd-EOB-DTPA-enhanced MRI taken before treatment. As a result, in cases with high signal nodules in the hepatobiliary phase (*n* = 8), the median PFS was 2.7 months, whereas in cases with low signal nodules (*n* = 10), it was 5.8 months (*p* = 0.007). There was also a significant difference in the period until tumor enlargement, indicating that the hepatobiliary phase of Gd-EOB-DTPA-enhanced MRI is a promising imaging biomarker to predict the therapeutic effect of anti-PD-1/PD-L1 monotherapy [83].

Moreover, it should be noted that multiple studies have reported that the efficacy of lenvatinib treatment for HCC was not affected by the signal intensity of the hepatobiliary phase of EOB-MRI [84,85].

Murai et al. focused on increasingly prevalent non-viral HCC cases. They extracted genomic DNA and total RNA from tumor tissues for profiling and then compared them with pathological findings to identify sensitivity to immunotherapy. Steatotic HCC accounted for 23% of non-viral HCC cases, which showed an immune-rich, yet immune-exhausted, tumor immune microenvironment characterized by T cell exhaustion, infiltration of M2 macrophages and cancer-associated fibroblasts (CAFs), high expression of immune PD-L1, and activation of TGF-β signaling. Histological fatty deposition of resected HCC tissue and Fat Fraction Corrected for Spectral Complexity and Inhomogeneities (FFCSI) measured by MRI were strongly correlated. The retrospective review of 30 HCC patients evaluated by MRI before atezolizumab–bevacizumab combination therapy confirmed a significantly longer PFS in patients with steatotic HCC [86].

Whether MRI findings are useful as imaging biomarkers to predict the efficacy of systemic therapy for HCC remains to be demonstrated with prospective validation studies. However, as the number of cohorts of systemic therapy using ICIs, such as atezolizumab–bevacizumab combination therapy, increases in clinical practice, we expect a consensus to be formed.

### 4.4. Problems with Wnt/β-Catenin Mutations as a Biomarker and MRI Findings as Imaging Biomarkers

Sasaki et al. reported that HCC patients receiving combination therapy of atezolizumab and bevacizumab with a high-intensity EOB-MRI hepatobiliary phase suggesting Wnt/β-catenin signal activation, had a shorter PFS than the low-intensity HCC patients in the atezolizumab + bevacizumab group. This MRI finding was not associated with the treatment effect of lenvatinib [87]. However, in a study by Kuwano et al. that used pretreatment tumor biopsies rather than EOB-MRI, there was no significant difference in the treatment effect or PFS of those receiving atezolizumab/bevacizumab combination therapy that depended on the presence or absence of Wnt/β-catenin activation [88]. These discrepancies may be caused by biases resulting from each study being retrospective and having a small number of cases, but other research results suggest otherwise.

Previous research results indicated there might be a discrepancy between EOB-MRI hepatocellular phase uptake findings and Wnt/β-catenin mutations. The transcription factor HNF4α maintains mature hepatocyte function and was decreased in dedifferentiated HCC, leading to the decreased expression of OATP1B3 regardless of Wnt/β-catenin mutations. This resulted in a loss of gadoxetic acid uptake in the hepatocyte phase, which may cause a mismatch [89,90].

Another question is whether the presence of Wnt/β-catenin mutations in HCC always results in a suppressed immune response. Sia et al. reported that about 25% of HCC cases had a subtype of an immune class characterized by immune activation, with overexpression of adaptive immune response genes, such as CD8A, CD3E, IFNG, CXCL9, and others, termed the active immune response subtype. Additionally, immunosuppressive signals, including TGF-β, and M2 macrophages were present in the exhausted immune response subtype. The authors also stated that a better response to ICI treatment was expected for this immune class [91].

This research group further investigated the immune characteristics of HCC cases outside this immune class. They found that about 10% of HCC cases had an immune-like class characterized by high IFN signaling, cytokines, and a diverse T cell repertoire, despite the significant activation of Wnt/β-catenin signaling by CTNNB1 mutation. This led them to classify HCC into an inflamed class, which includes the immune class and the immune-like class, as well as other non-inflamed classes. They suggested an “inflamed signature” consisting of 20 genes that accurately indicated the inflamed class and confirmed a significant overexpression of this signature in a group of patients who showed a partial response (PR) with ICI treatment in an external cohort compared with a group of patients who had stable disease (SD) or progressive disease (PD) [92].

When evaluating the tumor microenvironment, immunostaining in the inflamed class showed enrichment of intratumoral CD8+ T cells (CD8 ≥ 1%, 58% vs. 30%, *p* = 0.08) and PD-L1 (PD-L1 ≥ 1%, 21% vs. 4%, *p* = 0.19) compared with the non-inflamed class. Additionally, an analysis using CIBERSORT, which estimates the presence and ratios of immune cell subsets within tissues from gene expression data, showed a significantly higher proportion of CD8+ T cells (*p* = 3.51 × 10^−7^) and M1 macrophages (*p* = 1.82 × 10^−4^). However, in the immune-like class, M2 macrophages were significantly excluded (*p* = 1.78 × 10^−6^) [92].

Furthermore, the authors created a 13-protein signature as a liquid biopsy-based biomarker to identify the inflamed class using a cohort with blood samples. They suggested that when treating HCC, whether to distinguish between the inflamed and non-inflamed classes using the 20-gene signature in tumor tissues or the liquid biopsy-based signature should be considered [92].

Interestingly, a study using a dataset of over 9000 solid cancer cases across 31 types from TCGA found that activation of the Wnt/β-catenin pathway was often associated with reduced T cell infiltration in most human cancers. A significant inverse correlation was observed between β-catenin protein levels and T cell inflammatory gene expression in 177 HCC cases [80]. According to this study, the degree of the inverse correlation between β-catenin protein levels and T cell inflammatory gene expression varied by cancer type. Furthermore, in certain types of cancer, such as colorectal and rectal cancers, examples where T cell inflammation and activation of the Wnt/β-catenin pathway coexist are common [80]. This suggests that although activation of the Wnt/β-catenin pathway often hinders T cell inflammation in human cancers, this is not always the case. However, it appears that subclasses, such as the immune-like class, have not been proposed for other types of cancers.

Several studies have suggested that the hepatocyte phase of EOB-MRI may not match Wnt/β-catenin mutations and that some HCC cases of the immune-like class do not suppress immune responses against tumors, even when they have Wnt/β-catenin mutations. Therefore, findings from the hepatobiliary phase of EOB-MRI might be useful as biomarkers to predict the therapeutic effect of ICIs or the combination therapy of atezolizumab and bevacizumab. However, these findings do not necessarily indicate the presence or absence of Wnt/β-catenin mutations. When using the presence or absence of Wnt/β-catenin mutations as biomarkers to predict the therapeutic effect of ICIs, it is necessary to consider the existence of the immune-like class.

### 4.5. Potential Biomarkers to Predict the Therapeutic Effect of ICI Therapy

Spahn et al. reported that baseline levels of AFP 400 μg/L at the beginning of ICI therapy (anti-PD-1 antibodies) were linked to higher rates of PR or CR as the best responses and lower rates of PD. Furthermore, AFP levels below 400 μg/L were linked to considerably prolonged PFS and OS [93]. It was also reported that early AFP decline was associated with a favorable response to ICI therapy [94]. This report suggests that early AFP level changes are important biomarkers of ICI therapy efficacy.

Scheiner et al. created a training set of 190 cases and a validation set of 102 cases from a database of HCC cases in Europe that had received PD-L1/PD-1-based immunotherapy. Seventy-five cases (40%) in the training set and 25 cases (25%) in the validation set were patients who had received atezolizumab and bevacizumab combination therapy. In the training set, the investigated baseline parameters were etiology, whether immunotherapy was primary or after other treatments, Child–Pugh class, Eastern Cooperative Oncology Group performance status, radiological criteria, including the presence of major vessel invasion and extrahepatic metastasis, and serum AFP and C-reactive protein (CRP) levels. Serum AFP < 100 vs. ≥100 ng/mL and CRP < 1 vs. ≥1 mg/dL were identified as independent prognostic factors in a multivariate analysis, and the CRAFITY score was developed using these values [95].

Patients with a score of 0 (CRAFITY-low: AFP < 100 ng/mL and CRP < 1 mg/dL) had the longest OS, followed by those with a score of 1 (CRAFITY-intermediate: either AFP ≥ 100 ng/mL or CRP ≥ 1 mg/dL), and those with a score of 2 (CRAFITY-high: both AFP ≥ 100 ng/mL and CRP ≥ 1 mg/dL). Similarly, the best treatment effect was seen in patients with a low CRAFITY score. This study also validated a cohort of 204 cases of sorafenib administration. The CRAFITY score was associated with the survival of the individuals, but not with the therapeutic effect [95]. C-statistics, a statistical indicator that was used to evaluate the performance and predictive ability of the model in the CRAFITY score, was 0.62 for the derivation and validation cohorts. Although not highly accurate, it is very simple to use in routine practice and may be useful when predicting responses to ICI.

In a multi-institutional retrospective study in Japan, the CRAFITY score of 297 patients who received atezolizumab and bevacizumab combination therapy was analyzed. The median PFS in the CRAFITY score 0, 1, and 2 groups was 11.8, 6.5, and 3.2 months, respectively (*p* < 0.001). The median OS in patients with CRAFITY scores of 0, 1, or 2 was not reached, 14.3 months, and 11.6 months, respectively. This study showed the CRAFITY score might be useful for predicting therapeutic outcomes [96].

The pre-treatment NLR reflects the inflammatory response to cancer and was reportedly associated with patient prognosis and response to ICI treatment in various tumors [97,98,99,100,101]. Eso et al. analyzed the course of 40 HCC patients who received atezolizumab and bevacizumab combination therapy and found that the NLR value was significantly lower in the complete response (CR), PR, and SD groups than in the PD group (2.47 vs. 4.48, *p* = 0.013). Using the optimal NLR cut-off value (3.21) determined by receiver operating characteristic curve analysis for predicting responses, they also found that patients with an NLR ≤ 3.21 had significantly better PFS than patients with an NLR > 3.21 [101].

A similar examination was conducted in a multi-institutional joint study in Japan, where the cumulative OS rate was significantly different between patients with low NLR (<3.0) and high NLR (≥3.0) (*p* = 0.001). Conversely, there was no difference in the cumulative PFS or response between patients with low and high NLR values. In Cox proportional hazard modeling analysis using inverse probability weighting, an NLR of at least 3.0 was significantly associated with OS [102].

Because the CRAFITY score and NLR are parameters that can be easily obtained from blood samples, it is necessary to rigorously validate whether they should be used as biomarkers in routine practice.

Myojin et al. measured the levels of 34 baseline plasma proteins in patients with advanced HCC who received atezolizumab + bevacizumab therapy and found that plasma IL-6 levels were a significant predictor of non-response to this therapy. They confirmed that the PFS and OS were significantly shorter in the high IL-6 group than in the low IL-6 group [103].

Matsumae et al. evaluated pretreatment cfDNA in HCC patients with atezolizumab and bevacizumab. Patients with high cfDNA levels had a significantly lower ORR than those with low cfDNA levels. The PFS and OS were also significantly shorter in patients with high cfDNA levels than in patients with low cfDNA levels. Pretreatment cfDNA may be useful for predicting the therapeutic outcome of HCC patients treated with atezolizumab and bevacizumab [104]. Further investigation is needed to determine whether cfDNA concentration and composition affect immunotherapy responses to HCC.

Kim et al. investigated whether elevated levels of anti-drug antibodies in atezolizumab and bevacizumab affected therapeutic efficacy and T cell function. Patients with high ADA levels treated with atezolizumab and bevacizumab had a lower response rate and worse PFS and OS than patients with low ADA levels. Patients with high ADA levels had lower serum atezolizumab concentrations, reduced CD8-positive T cell proliferation, and lower IFN-γ and tumor necrosis factor-α production from CD8-positive T cells than patients with low ADA levels [105]. Therefore, elevated ADA levels after atezolizumab and bevacizumab administration may be considered a poor prognostic factor.

As will be discussed below, the results of the HIMALAYA trial have made it possible to administer a combination of durvalumab and tremelimumab as a first-line treatment. However, comparisons of treatment outcomes have indicated that the first-line treatment of choice for unresectable HCC cases is currently atezolizumab and bevacizumab combination therapy [106,107].

Several studies have reported gene expression profiles as biomarkers of responses to ICI therapy. Gene expression profiling of tissues revealed that PD-1 and PD-L1 expressions, biomarkers of inflammation (CD3 and CD8), and inflammatory gene signatures (CD274, CD8A, LAG3, STAT1) tended to be associated with improved survival and responses of HCC patients treated with anti-PD1 antibody [108]. Haber et al. reported that IFN signaling and major histocompatibility complex-related genes were key molecular features of HCCs that responded to anti-PD1 [109].

As mentioned previously, Zhu et al. used transcriptome analysis to derive an atezolizumab + bevacizumab response signature (ABRS) comprised of 10 genes associated with a response to atezolizumab + bevacizumab (defined as CR or PR). High expression of the ABRS, as well as the existing immune gene CD274 (PD-L1 mRNA) or the Teff sign (CXCL9, PRFI, and GZMB), were associated with longer PFS in patients treated with atezolizumab + bevacizumab. The Treg signature (CCR8, BATF, CTSC, TNFRSF4, FOXP3, TNFRSF18, IKZF2, and IL2RA) was also related to improved PFS and OS when the Treg/Teff signature ratio was low in the IMbrave150 study, which compared atezolizumab + bevacizumab with sorafenib treatment [72].

Multiplex IHC analysis in this study showed that in GO30140 cohort A, responding patients (CR/PR) had a higher density of infiltrating CD8+ T cells, CD3+ T cells, and GZMB+/CD3+ T cells in tumor areas than non-responders (SD/PD). The density of tumor-infiltrating CD8+ T cells in IMbrave150 baseline tumor samples was analyzed and patients with a high density of tumor-infiltrating CD8+ T cells (defined by the split median) had significantly longer OS when treated with atezolizumab + bevacizumab combination therapy compared with those treated with sorafenib [73]. These studies highlight an important point: the state of T cell immunity in the tumor microenvironment before treatment ultimately influences the treatment effect of atezolizumab + bevacizumab combination therapy for HCC.

Combination therapy with the anti-PD-L1 antibody durvalumab with anti-CTLA-4 antibody tremelimumab for unresectable HCC cases has surpassed the control drug sorafenib in the HIMALAYA trial and has been approved as a first-line therapy [15]. Among the factors studied as potential biomarkers in the HIMALAYA trial, the only results currently available relate to the PD-L1 status of the tumor prior to therapy. According to the subgroup analysis, there was no difference in benefit for the combination of two ICIs compared with sorafenib, regardless of whether PD-L1 expression was positive or negative [15].

Kuwano et al. examined the relationship between the tumor infiltration of CD8+ T cells detected by IHC staining of liver tumor biopsies before treatment initiation and the therapeutic effect of drug therapy. In cases with a high level of CD8+ T cell tumor infiltration, the PFS of patients treated with atezolizumab + bevacizumab combination therapy was significantly extended, and the response rate was also significantly improved compared with cases with low levels. However, in patients receiving lenvatinib, there was no association between CD8+ T cell tumor infiltration and the response rate or PFS [110]. Although this study included a limited number of patients, it suggests that evaluating CD8+ T cell tumor infiltration alone, without any transcriptomics analysis or genomic profiling, may serve as a useful biomarker to help decide the treatment method choice and predict the therapeutic response to drug therapy in HCC.

Another interesting finding was that the combination therapy showed clear advantages over sorafenib treatment in cases of HBV-related and non-viral HCC, but not in cases of HCV-related HCC. A similar trend was observed with durvalumab monotherapy [15]. As mentioned in the sorafenib section of this review, this may be because sorafenib has greater benefits for HCV-related HCC [18]. Nevertheless, there is no clear biomarker candidate for the durvalumab and tremelimumab combination therapy. The CRAFITY score and NLR can be easily validated, but to date, no reports have confirmed their effectiveness with this treatment.

Lee et al. reported a relationship between microbiota and the efficacy of ICIs for HCC patients. Lachnospiraceae and Veillonellaceae were enriched in the feces of patients with OR. In contrast, apparent increases in Prevotellaceae and Enterobacteriaceae, but a reduced abundance of Lachnospiraceae and Veillonellaceae, were observed in patients with PD after immunotherapy [111]. Spahn et al. revealed systemic antibiotics were associated with worse outcomes in HCC patients undergoing anti-PD-1 treatment [93]. This result indicates that gut dysbiosis adversely affects HCC ICI therapy. The composition of gut microbiota might be a biomarker for the therapeutic efficacy of immunotherapy for patients with HCC and may be targeted as a treatment to enhance the efficacy of ICI therapy.

ICI therapy often induces inflammatory reactions termed irAEs, which can affect various organs, including the skin, gastrointestinal, liver, respiratory, thyroid, and pituitary glands, and occur at any time. Recently, several studies reported that irAEs were associated with the efficacy of ICI therapy in patients with HCC [112]. Fukushima et al. reported that low-grade irAEs were strongly correlated with the PFS and OS in HCC patients treated with atezolizumab and bevacizumab [113]. IrAEs, particularly low-grade irAEs, are markers that predict better ICI therapy efficiency in HCC patients.

**Table 2 cancers-15-04345-t002:** Factors Influencing Prognosis/Efficacy in Hepatocellular Carcinoma with Immune Checkpoint Inhibitors.

Therapeutics	Study Design	Number of Cases	Prognostic and Predictive Factors	Outcome	Statistical Analysis HR (95% CI)	*p*-Value	Author [Reference No.]
Anti-PD-(L)1-based immunotherapy	Meta-analyses of 3 phase 3 trials: Checkmate 459 (nivolumab vs. sorafenib), IMbrave 150 (Atezo/Beva vs. sorafenib), KEYNOTE-240 (Pembrolizumab vs. Placebo)	ICI 985 Nivolumab 371 Pembrolizumab 278 Atezo/Beva 336 Control 672 Sorafenib 372 + 165 Placebo 135	HBV	OS ↑	Univariate 0.64 (0.49–0.83)	0.0008	Pfister D et. al. [75]
			HCV	OS ↑	Univariate 0.68 (0.48–0.97)	0.04	
	Retrospective (ICI single arm)	exploratory cohort 130 validation cohort 118	NAFLD	OS ↓	Multivariate 2.6. (1.2–5.6)	0.017	
Atezo/Beva Lenvatinib (Sorafenib)	Retrospective	Non-viral cohort Atezo/Beva 190 Lenvatinib 569	Lenvatinib	OS ↑	Multivariate 0.65 (0.44–0.95)	0.0268	Rimini M et al. [77]
				PFS ↑	Multivariate 0.67 (0.51–0.86)	0.035	
		NAFLD/NASH cohort Atezo/Beva 82 Lenvatinib 254	Lenvatinib	OS ↑	Multivariate 0.46 (0.26–0.84)	0.011	
				PFS ↑	Multivariate 0.55 (0.38–0.82)	0.031	
Anti-PD-(L)1 monotherapy	Retrospective, single arm	18	Hyperintensity tumor (RER ‡ ≥ 0.9) on EOB-MRI	PFS ↓	Multivariate 7.78 (1.59–38.1)	0.011	Aoki T et. al. [83]
Atezo/Beva	Retrospective validation based on multiomics study, single arm	Non-viral HCC 30	Steatotic HCC	PFS ↑	Univariate	<0.05	Murai H et.al. [86]
Atezo/Beva Lenvatinib	Retrospective, separate single arm (not vs. lenvatinib)	Atezo/Beva 35	Heterogeneous tumor on EOB-MRI	PFS ↓	Univariate	0.007	Sasaki R et.al. [87]
			Hyperintensity tumor (RER ‡ ≥ 0.9) on EOB-MRI	PFS ↓	Univariate	0.012	
		Lenvatinib 33	(no significant factor)		-		
Anti-PD-(L)1-based immunotherapy	Retrospective, single arm	24	20 gene inflamed signature (CCL5, CD2, CD3D, CD48, CD52, CD53, CXCL9, CXCR4, FYB, GZMA, GZMB, GZMK, IGHG1, IGHG3, LAPTM5, LCP2, PTPRC, SLA, TRAC, TRBC2)	PR ↑	Wilcoxon rank sum	0.047	Montironi C et.al. [92]
Anti-PD-1 monotherapy	Retrospective, single arm	99	AFP < 400	OS ↑	Univariate 2.81 (1.56–4.97)	<0.0001	Spahn et. al. [93]
				PFS ↑	Univariate 1.33 (0.86–2.06)	<0.05	
Anti-PD-1 monotherapy	Retrospective, single arm	60	AFP response as a >20% decline	OS ↑	Univariate 0.09 (0.02–0.44)	<0.001	Shao et. al. [94]
				PFS ↑	Univariate 0.13 (0.04–0.39)	0.003	
Anti-PD-(L)1-based immunotherapy Sorafenib	Retrospective, separate single arm (not vs. sorafenib)	Anti-PD-(L)1-based immunotherapy: training cohort 190 (anti-PD-(L)1 mono 110, Atezo/Beva 75, Others 5) validation cohort 102 (anti-PD-(L)1 mono 68, Atezo/Beva 25, Anti-PD-(L)1 + TKI 7, Others 2)	Child–Pugh A	OS ↑	Multivariate 2.3 (1.5–3.4)	<0.001	Scheiner B et.al. [95]
			ECOG PS 0	OS ↑	Multivariate 2.1 (1.4–3.2)	<0.001	
			AFP < 100	OS ↑	Multivariate 1.7 (1.2–2.6)	0.007	
			CRP < 1	OS ↑	Multivariate 1.7 (1.2–2.6)	0.007	
			CRAFITY score †	OS	Univariate	0.001	
			CRAFITY low		1		
			CRAFITY int.		2.0 (1.1–3.4)		
			CRAFITY high		3.6 (2.1–6.2)		
			CRAFITY score †	ORR	Chi-square	0.001	
				DCR	Chi-square	<0.001	
			CRAFITY score †	OS	Univariate	0.001	
				DCR	Chi-square	0.037	
		Sorafenib 204	CRAFITY score †	OS	Univariate	<0.001	
Ate/Bev	Retrospective, single arm	297	AFP < 100	PFS ↑	Multivariate	<0.001	Hatanaka T et.al. [96]
				OS ↑	Multivariate	0.028	
			CRP < 1	PFS ↑	Multivariate	<0.001	
				OS ↑	Multivariate	0.032	
			CRAFITY score †	PFS	Univariate	<0.001	
				OS	Univariate		
				DCR	Chi-square	0.029	
Ate/Bev	Retrospective, single arm	40	NLR ≤ 3.21	PFS ↑	Univariate	<0.0001	Eso Y et.al [101]
Ate/Bev	Retrospective, single arm	249	NLR > 3	OS ↓	Multivariate 3.37 (1.02–11.08)	0.001	Tada T et.al. [102]
Atezo/Beva Sorafenib	Retrospective pooled analysis of the phase 1b GO30140 (single arm) and the phase 3 trial IMbrave 150 (Atezo/Beva vs. sorafenib)	GO30140 arm A cohort Atezo/Beva 90 (single arm)	**<Transcriptome analyses>**				Zhu AX et. al. [72]
			ABRS ^a^ high	PFS ↑	Univariate 0.51 (0.3–0.87)	0.013	
			CD274 ^b^ high	PFS ↑	Univariate 0.42 (0.25–0.72)	0.0011	
			Teff ^c^ high	PFS ↑	Univariate 0.46 (0.27–0.78)	0.0035	
			**<In situ analyses>**				
			CD8+ T cell density	CR/PR ↑	Student T	0.007	
			CD3+ T cell density	CR/PR ↑	Student T	0.039	
			CD3+ GZMB + T cell density	CR/PR ↑	Student T	0.044	
			MHC1 + tumor cells	CR/PR ↑	Student T	0.0087	
		IMbrave 150 (Atezo/Beva 119 sorafenib 58)	**<Transcriptome analyses>**				
			ABRS ^a^ high	PFS ↑	Multivariate 0.49 (0.25–0.97)	0.041	
				OS ↑	Multivariate 0.26 (0.11–0.58)	0.0012	
			CD274 ^b^ high	PFS ↑	Multivariate 0.46 (0.25–0.86)	0.015	
				OS ↑	Multivariate 0.3 (0.14–0.64)	0.002	
			Teff ^c^ high	PFS ↑	Multivariate 0.52 (0.28–0.99)	0.047	
				OS ↑	Multivariate 0.24 (0.11–0.5)	0.0002	
			Treg ^d^/Teff ^c^ low	PFS ↑	Multivariate 0.42 (0.22–0.79)	0.007	
				OS ↑	Multivariate 0.24 (0.11–0.54)	0.0006	
			GPC3 low	PFS ↑	Multivariate 0.47 (0.27–0.81)	0.006	
				OS ↑	Multivariate 0.29 (0.13–0.62)	0.002	
			AFP low	PFS ↑	Multivariate 0.49 (0.28–0.87)	0.014	
				OS ↑	Multivariate 0.32 (0.14–0.73)	0.007	
			**<In situ analyses>**				
			CD8+ T cell high dens.	OS ↑	Multivariate 0.29 (0.14–0.61)	0.0011	
				PFS ↑	Multivariate 0.54 (0.29–1.00)	0.053	
			**<Genetic profiling>**				
			CTNNB1 WT	OS ↑	Multivariate 0.42 (0.19–0.91)	3 × 10^−4^	
				PFS ↑	Multivariate 0.45 (0.27–0.86)	0.0086	
			TERT Mut	OS ↑	Multivariate 0.38 (0.16–0.89)	7.8 × 10^−5^	
				PFS ↑	Multivariate 0.61 (0.33–1.10)	0.047	
Atezo/Beva	Retrospective, single arm	34	High plasma IL-6	PFS ↑	Univariate	<0.05	Myojin Y et.al. [103]
					Multivariate 2.785 (1.216–6.38)	0.01	
				OS ↑	Univariate	<0.05	
Atezo/Beva Lenvatinib	Retrospective, separate single arm (not vs. lenvatinib)	Atezo/Beva 24	High-level CD8+ TILs	PFS ↑	Univariate	0.041	Kuwano A et.al. [110]
				ORR ↑	Chi-square	0.012	
				DCR ↑	Chi-square	0.031	
		Lenvatinib 15	(No significant factor)				
Anti-PD-1 monotherapy	CheckMate 040 trial (nivolumab)	37	**<Genetic profiling>** Inflammatory gene signatures (CD274, CD8A, LAG3, STAT1)	OS ↑	Univariate	0.01	Sangro et.al. [108]
			PD-L1	OS ↑	Univariate	0.032	
Anti-PD-1 monotherapy	Retrospective, single arm	99	Antibiotic treatment	PD ↑	Chi-square	<0.05	Spahn et.al. [93]
				PFS ↓	Univariate 1.65 (0.9–3.0)	<0.05	
Atezo/Beva	Retrospective, single arm	85	Cell free DNA low	OS ↑	Univariate	0.018	Matsumae et.al [104]
				PFS ↑	Univariate	0.021	
Atezo/Beva	Retrospective, single arm	174	Anti-drug antibodies	OS ↓	Univariate 5.81 (2.7–12.5)	0.001	Kim et.al. [105]
				PFS ↓	Univariate 2.52 (1.27–5.01)	0.006	
Atezo/Beva	Retrospective, single arm	150	Grade 1/2 irAEs	OS ↑	Multivariate 0.09 (0.01–0.64)	0.017	Fukushima ey.al. [113]
				PFS ↑	Multivariate 0.34 (0.17–0.69)	0.003	

HR, hazard ratio; CI, confidence interval; PFS, progression-free survival; OS, overall survival; PD-1, programmed death receptor 1; PD-L1, programmed cell death ligand 1; HBV, hepatitis B virus; HCV, hepatitis C virus; NAFLD, nonalcoholic fatty liver disease; NASH, nonalcoholic steatohepatitis; RER, relative enhancement ratio; EOB-MRI, gadolinium ethoxybenzyl diethylenetriaminepentaacetic acid-enhanced magnetic resonance imaging; HCC, hepatocellular carcinoma; PR, partial response; ECOG, Eastern Cooperative Oncology Group; AFP, α-fetoprotein; CRP, C-reactive protein; ORR, overall response rate; DCR, disease control rate; NLR, neutrophil to lymphocyte ratio; CR, complete response; irAE, immune-related adverse events. Note: In the “Statistical analysis” section, “univariate” typically refers to the Kaplan–Meier method and log-rank test, and the inclusion of HR indicates the use of Cox regression. Additionally, “multivariate” typically refers to the utilization of the multivariate Cox regression model. ‡: (nodule SI/parenchyma SI on hepatobiliary phase images)/(nodule SI/parenchyma SI on precontract images). SI: signal intensity. †: CRAFITY-low: AFP < 100 and CRP < 1, intermediate: AFP ≥ 100 ng/mL or CRP ≥ 1 mg/dL, high: AFP ≥ 100 ng/mL and CRP ≥ 1 mg/dL. ^a^: ABRS, atezolizumab + bevacizumab response signature (including CXCR2P1, ICOS, TIMD4, CTLA4, PAX5, KLRC3, FCRL3, AIM2, GBP5, and CCL4). ^b^: CD274, PD-L1 mRNA. ^c^: Teff, T effector (including CXCL9, PRF1, and GZMB). ^d^: Treg, T regulatory (including CCR8, BATF, CTSC, TNFRSF4, FOXP3, TNFRSF18, IKZF2, and IL2RA).

## 5. Conclusions and Future Directions

In this review, we discussed published research findings on biomarkers to predict the therapeutic effects of drugs available for unresectable HCC tumors. As TKIs will continue to be used as secondary therapies, the search for biomarkers must continue. Among the factors mentioned, hepatic function, underlying hepatic disease, and the NLR may serve as vague indications of utility, but they are far from being decisive in drug selection. Because performing prospective trials of existing TKIs might be difficult in the future, we think it is worthwhile to proactively investigate whether biomarker candidates including DKK-1, ST6GAL1, and regorafenib have predictive value, because they have been indicated in several studies. We encourage this investigation within existing cohorts that have retained blood samples.

Prognostic and predictive factors for ICI therapy are listed in Table 3. For the current standard treatment of atezolizumab and bevacizumab combination therapy, routine examinations such as underlying hepatic disease, CRAFITY score, and the NLR seem to provide some guidance. Furthermore, imaging diagnostics, such as the evaluation of fat deposition throughout the EOB-MRI hepatobiliary phase or FFCSI and information from MRI examinations, can be useful. At present, the most important factor is thought to be an accurate assessment of the state of T cell immunity in the tumor microenvironment prior to treatment. The literature suggests that evaluating the tumor microenvironment and immune environment is more achievable compared with evaluating the diversity and heterogeneity of tumor cells. Assessing CD8+ T cell infiltration by collecting tumor tissues can be performed relatively easily, even in general hospitals. Thus, this has a high potential to become a practical biomarker.

Currently, there are no approved biomarkers, except for AFP and ramucirumab, for the prediction of response/resistance to systemic therapy using ICIs and TKIs. In general hospitals that treat HCC patients, it is common to administer drug therapy for advanced HCC cases based on guidelines informed solely by the results of blood tests and computed tomography scans. Although this approach is not inherently flawed, based on our perspective, gathering diagnostic materials such as blood and tissue samples, coupled with performing MRI image evaluations before treatment, can offer deeper insights into the state within the tumor tissue.

## Figures and Tables

**Table 3 cancers-15-04345-t003:** Potential predictive biomarkers for treating hepatocellular carcinoma with immune checkpoint inhibitors.

Prognostic and Predictive Factors	Therapeutics	Study Design	Outcome	Statistical Analysis HR (95% CI)	*p*-Value	Author [Reference No.]
<Etiology>						
HBV	Anti-PD-(L)1-based immunotherapy	Meta-analyses of 3 phase 3 trials: Checkmate 459 (nivolumab vs. sorafenib), IMbrave 150 (Atezo/Beva vs. sorafenib), KEYNOTE-240 (Pembrolizumab vs. placebo)	OS	Univariate 0.64 (0.49–0.83)	0.0008	Pfister D et al. [75]
HCV			OS	Univariate 0.68 (0.48–0.97)	0.04	
NAFLD		Retrospective (ICI single arm)	OS	Multivariate 2.6 (1.2–5.6)	0.017	
**<Liver function and** **general condition>**						
Child–Pugh A	Anti-PD-(L)1-based immunotherapy	Retrospective, separate single arm (not vs. sorafenib)	OS	Multivariate 2.3 (1.5–3.4)	<0.001	Scheiner B et al. [95]
ECOG PS0			OS	Multivariate 2.1 (1.4–3.2)	<0.001	
**<Image>**						
hypertensive tumor (RER ‡ ≥ 0.9) on EOB-MRI	Anti-PD-(L)1 monotherapy	Retrospective, single arm	PFS	Multivariate 7.78 (1.59–38.1)	0.011	Aoki T et al. [83]
	Atezo/Beva, lenvatinib	Retrospective, separate single arm (not vs. lenvatinib)	PFS	Univariate	0.012	Sasaki R et al. [87]
Heterogenous tumor on EOB-MRI			PFS	Univariate	0.007	
Steatotic HCC	Atezo/Beva	Retrospective validation based on multiomics study, single arm	PFS	Univariate	<0.05	Murai H et al. [86]
**<Blood marker>**						
AFP(<400)	Anti-PD-1 monotherapy	Retrospective, single arm	OS	Univariate 2.81 (1.56–4.97)	<0.0001	Spahn et. al. [93]
			PFS	Univariate 1.33 (0.86–2.06)	<0.05	
AFP response as a >20% decline	Anti-PD-1 monotherapy	Retrospective, single arm	OS	Univariate 0.09 (0.02–0.44)	<0.001	
			PFS	Univariate 0.13 (0.04–0.39)	0.003	
AFP (<100)	Anti-PD-(L)1-based immunotherapy	Retrospective, separate single arm	OS	Multivariate 1.7 (1.2–2.6)	0.007	Scheiner B et al. [95]
	Atezo/Beva	Retrospective, single arm	PFS	Multivariate	<0.001	Hatanaka T et.al. [96]
			OS	Multivariate	0.028	
CRP (<1)	Anti-PD-(L)1-based immunotherapy	Retrospective, single arm	OS	Multivariate 1.7 (1.2–2.6)	0.007	Scheiner B et.al. [95]
	Atezo/Beva	Retrospective, single arm	PFS	Multivariate	<0.001	Hatanaka T et.al. [96]
			OS	Multivariate	0.032	
CAFITY score †	Anti-PD-(L)1-based immunotherapy	Retrospective, separate single arm (not vs. sorafenib)	OS	Univariate	0.001	Scheiner B et.al. [95]
CAFITY low				1		
CAFITY int				2.0 (1.1–3.4)		
CAFITY high				3.6 (2.1–6.2)		
CAFITY score †			ORR	Chi-square	0.001	
			DCR	Chi-square	<0.001	
CAFITY score †	Atezo/Beva	Retrospective, single arm	PFS	Univariate	<0.001	Hatanaka T et.al. [96]
			OS	Univariate		
			DCR	Chi-square	0.029	
NLR (>3.21)	Atezo/Beva	Retrospective, single arm	PFS	Univariate	<0.0001	Eso Y et.al [101]
NLR (>3)	Atezo/Beva	Retrospective, single arm	OS	Multivariate 3.37 (1.02–11.08)	0.001	Tada T et.al. [102]
IL-6	Atezo/Beva	Retrospective, single arm	PFS	Univariate	<0.05	Myojin Y et.al. [103]
				Multivariate 2.785 (1.216–6.38)	0.01	
			OS	Univariate	<0.05	
**<Transcriptome>**						
ABRS ^a^	Atezo/Beva (not vs. sorafenib)	Retrospective analysis of GO30140 arm A cohort	PFS	Univariate 0.51 (0.3–0.87)	0.013	Zhu AX et. al. [72]
	Atezo/Beva vs. sorafenib	Retrospective analysis of IMbrave 150	PFS	Multivariate 0.49 (0.25–0.97)	0.041	
			OS	Multivariate 0.26 (0.11–0.58)	0.0012	
CD274 ^b^	Atezo/Beva (not vs. sorafenib)	Retrospective analysis of GO30140 arm A cohort	PFS	Univariate 0.42 (0.25–0.72)	0.0011	
	Atezo/Beva vs. sorafenib	Retrospective analysis of IMbrave 150	PFS	Multivariate 0.46 (0.25–0.86)	0.015	
			OS	Multivariate 0.3 (0.14–0.64)	0.002	
Teff ^c^	Atezo/Beva (not vs. sorafenib)	Retrospective analysis of GO30140 arm A cohort	PFS	Univariate 0.46 (0.27–0.78)	0.0035	
	Atezo/Beva vs. sorafenib	Retrospective analysis of IMbrave 150	PFS	Multivariate 0.52 (0.28–0.99)	0.047	
			OS	Multivariate 0.24 (0.11–0.5)	0.0002	
Treg ^d^/Teff ^c^			PFS	Multivariate 0.52 (0.28–0.99)	0.047	
			OS	Multivariate 0.24 (0.11–0.5)	0.0002	
GPC3			PFS	Multivariate 0.47 (0.27–0.81)	0.007	
			OS	Multivariate 0.29 (0.13–0.62)	0.002	
AFP			PFS	Multivariate 0.49 (0.28–0.87)	0.014	
			OS	Multivariate 0.32 (0.14–0.73)	0.007	
**<In situ marker>**						
CD8+ T cell	Atezo/Beva (not vs. sorafenib)	Retrospective analysis of GO30140 arm A cohort	CR/PR	Student T	0.007	Zhu AX et al. [72]
CD3+ T cell			CR/PR	Student T	0.039	
CD3 + GZMB + T cell			CR/PR	Student T	0.044	
MHC1 + tumor cells			CR/PR	Student T	0.0087	
**<Genetic marker>**						
CTNNB1 WT	Atezo/Beva vs. sorafenib	Retrospective analysis of IMbrave 150	OS	Multivariate 0.42 (0.19–0.91)	3 × 10^−4^	Zhu AX et al. [72]
			PFS	multivariate 0.45 (0.27–0.86)	0.0086	
TERT Mut			OS	multivariate 0.38 (0.16–0.89)	7.8 × 10^−5^	
			PFS	multivariate 0.61 (0.33–1.10)	0.047	
CD274, CD8A, LAG3, STAT1	Anti-PD-1 monotherapy	Retrospective analysis of CheckMate 040 trial	OS	Univariate	0.01	Sangro et al. [108]
PD-L1			OS	Univariate	0.032	
**<Other>**						
Cell-free DNA	Anti-PD-1 monotherapy	Retrospective, single arm	OS	Univariate	0.018	Matsumae et.al. [104]
			PFS	Univariate	0.021	

HR, hazard ratio; CI, confidence interval; PFS, progression-free survival; OS, overall survival; PD-1, programmed death receptor 1; PD-L1, Programmed cell death ligand 1; HBV, hepatitis B virus; HCV, hepatitis C virus; NAFLD, nonalcoholic fatty liver disease; NASH, nonalcoholic steatohepatitis; RER, relative enhancement ratio; EOB-MRI, gadolinium ethoxybenzyl diethylenetriaminepentaacetic acid-enhanced magnetic resonance imaging; HCC, hepatocellular carcinoma; PR, partial response; ECOG, Eastern Cooperative Oncology Group; AFP, α-fetoprotein; CRP, C-reactive protein; ORR, overall response rate; DCR, disease control rate; NLR, neutrophil to lymphocyte ratio; CR, complete response; irAE, immune-related adverse events. Note: In the “Statistical analysis” section, “univariate” typically refers to the Kaplan–Meier method and log-rank test, and the inclusion of HR indicates the use of Cox regression. Additionally, “multivariate” typically refers to the utilization of the multivariate Cox regression model. ‡: (nodule SI/parenchyma SI on hepatobiliary phase images)/(nodule SI/parenchyma SI on precontract images). SI: signal intensity. †: CRAFITY-low: AFP < 100 and CRP < 1, intermediate: AFP ≥ 100 ng/mL or CRP ≥ 1 mg/dL, high: AFP ≥ 100 ng/mL and CRP ≥ 1 mg/dL. ^a^: ABRS, atezolizumab + bevacizumab response signature (including CXCR2P1, ICOS, TIMD4, CTLA4, PAX5, KLRC3, FCRL3, AIM2, GBP5, and CCL4). ^b^: CD274, PD-L1 mRNA. ^c^: Teff, T effector (including CXCL9, PRF1, and GZMB). ^d^: Treg, T regulatory (including CCR8, BATF, CTSC, TNFRSF4, FOXP3, TNFRSF18, IKZF2, and IL2RA).

## Data Availability

No new data were created or analyzed in this study. Data sharing is not applicable to this article.

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
