# Peer review of "Potential Predictive Biomarkers of Systemic Drug Therapy for Hepatocellular Carcinoma: Anticipated Usefulness in Clinical Practice"

_cancers, 2023, doi:10.3390/cancers15174345_

Round 1
Reviewer 1 Report
This is another comprehensive review on the systemic drug therapy for hepatocellular carcinoma. There have been quite a few of similar reviews in the literature. This present review will add to this pool of literature reviews. Nevertheless, the review is informative and a pleasure to read. The following suggestions could help to improve the presentation of the review:
The clarity of the tables presented in the text can be improved:
For examples, in Tale 1, Number of cases could be incorporated into Study design; in the Outcome column, it would be good to add upper arrow or down arrow to indicate increase or decrease survival; the statistical analysis could be incorporated into the HR column.
Similar presentation could be adopted for Tabel 2.
Lines 341 to 350 could not be located in the text.
To add something unique for this review, the authors could consider having a section entitled "Potential biomarkers for future patient selection" to incorporate sections 4.2-4.7.
Some structures of the sentences in the text need to be improved.
Reviewer 2 Report
In their paper Motomura et al. provided an overview of the investigated biomarkers for the prediction of the response to systemic therapies in hepatocellular carcinoma. Even if the topic is interesting and very timely, in this review there are several issues that deserve attention.
1. Title. I have not completely clear what the Authors mean with the terms “practical biomarkers”. Probably, they mean “biomarkers useful in clinical practice”. I also find that the term “Exploration” is not adequate in this title. Therefore, I suggest to change the all title of the paper.
2. Introduction. The introduction provided by the Authors tried to explain what are the currently approved systemic therapy options for patients with HCC, but it is a bit confusing. ICIs monotherapy (nivolumab, pembrolizumab) is not mentioned. Beyond the introduction, in which Authors should introduce the relevance of finding reliable predictive biomarkers (in order to better select the therapeutic approach for each patient), I suggest to include a paragraph explaining with more details the trials on systemic therapies conducted so far and the approved therapeutic options.
3. Aim. The Authors stated that “Therefore, in this review, we provide an overview of the wide range of research that has been conducted on HCC biomarkers from blood, tissue, or imaging information that can be used practically in general hospitals for predicting the therapeutic effect of systemic therapies before treatment begins.” (Page 2, lines 70-73). Their aim is to review the biomarkers that can be used in clinical practice for predicting the efficacy of systemic therapies. However, they should acknowledge in their paper that currently (except for AFP in the treatment with ramucirumab, which is the only example of biomarker-guided therapy) there are no approved biomarkers that can be used in clinical practice to select a therapy. Indeed, almost all the data they presented throughout their paper, despite promising in the future, at this time are purely interesting from a research point of view and need to be prospectively and extensively validated before the introduction in clinical practice (in several point of the manuscript, the Authors reported proof of concept studies in very small cohorts). Therefore, it is not correct to present the data included in the review as something that can be used in clinical practice and the sentence “information that can be used practically in general hospitals for predicting the therapeutic effect of systemic therapies before treatment begins” must be amended. Moreover, as the aim is to provide information about biomarkers that can be routinely used in general facilities, it should be noted that several results presented throughout the paper relate to techniques that are not immediately usable in general hospitals, as they require advanced technologies and dedicated personnel, are costly and time consuming (e.g., miRNA evaluation, cell cultures, identification of driver gene mutations).
4. In the presentation of results regarding biomarkers for prediction of effectiveness of both TKIs and ICIs, the paper is a bit confusing and not easy to follow (for instance, in the part of the manuscript describing biomarkers of response/resistance to ICIs there is not a linear presentation). Instead of presenting results divided by drug, I suggest to divide the paper in paragraphs according to the evaluated biomarker (e.g., for ICIs: markers of response/resistance in tumor genome [TMB, somatic mutations, gene expression profiling]; markers of response/resistance in tumor tissue [immunogenomic classification, PD-L1 expression, TILs]; markers of response/resistance associated to the host [etiology, microbiota]; circulating biomarkers…).
5. Especially in the part of the review regarding biomarkers of response/resistance to ICIs, there is some relevant literature that is missing:
- No mention has been made regarding gene expression profiling as biomarker of response/resistance (DOI: 10.1016/j.jhep.2020.07.026 and DOI: 10.1053/j.gastro.2022.09.005).
- Beyond CRAFITY score and NLR, other potential circulating biomarkers of response to ICIs (cfDNA, anti-drug antibodies, AFP) have been neglected and should be included in the manuscript.
6. In the paragraph dealing with the role of etiology on the response to treatment with ICIs, Authors should acknowledge that, etiology seems to have a role in modulating the response to ICIs, currently no recommendations regarding different treatments based on the underlying liver disease could be made. Indeed, all the available data derive from post-hoc analyses from RCTs and from retrospective studies. Moreover, other relevant points are: in RCTs there is a problem in defining “non-viral” HCCs and different studies include in this category different patients; trials with different drugs came to different conclusions regarding the efficacy of ICIs in non-viral HCC; not all the evidence are concordant in defining ICIs as less active in patients with non-viral etiology (see DOI: 10.1053/j.gastro.2023.02.042).
7. There are some data demonstrating that microbiota may have a role in modulating the response/resistance to ICIs. The Authors should include a paragraph regarding the potential role of microbiota as biomarker.
8. The development of immune-related adverse events also has been shown to be associated with response/resistance to ICIs, and Authors should include these data in their review (as clinical biomarkers to response to ICIs).
9. In their conclusions, the Authors stated that “we recommend conducting treatment after assessing the state within the tumor tissue as much as possible by collecting blood and tissue samples and performing MRI image evaluations before treatment.” It is unclear on what basis the authors make this claim. Such a recommendation could not be made based on data currently available. Instead, as mentioned in a previous comment, Authors should acknowledge that currently there is not approved biomarkers (except for the case of AFP and ramucirumab) for the prediction of response/resistance to systemic therapy (both ICIs and TKIs).
In my opinion, the English language of the manuscript requires revisions (in many parts I found the paper difficult to follow). I recommend the Authors to revise carefully all the manuscript.
Round 2
Reviewer 2 Report
I plaudit the efforts made by the authors to address my previous concerns and recommendations. I think that the new version of the manuscript is now improved.
I only have some other minor comments and suggestions:
- In the abstract, the term “immunocomplex” (line 19) is not appropriate. Please amend.
- In the introduction, the sentence “Additionally, cabozantinib, a TKI, was developed as a second-line or later therapy [15].” should be moved to the paragraph in which TKI therapy is presented (it is not appropriate and out of context to maintain this sentence after the presentation of ICIs therapy).
- The new paragraph included by the authors (4.8) include relevant information regarding predictive biomarkers. However, reporting such different information in the same paragraph (microbiota, ADAs, immune-related adverse effects, …) is extremely confusing. Therefore, I suggest integrating what is reported here in the previous paragraphs (e.g., ADAs in the paragraph where the circulating markers were discussed). Regarding the evaluated biomarkers in ICIs therapy, the authors can refer to the following paper for an example of how to organize the manuscript: DOI: 10.3390/biomedicines11041020
While improved over the original version, I feel there is ample room for further improvement in the English language.
